# Alterations of Oligodendrocyte and Myelin Energy Metabolism in Multiple Sclerosis

**DOI:** 10.3390/ijms241612912

**Published:** 2023-08-18

**Authors:** Eneritz López-Muguruza, Carlos Matute

**Affiliations:** 1Achucarro Basque Center for Neuroscience, 48940 Leioa, Spain; eneritz.lopez@ehu.eus; 2Department of Neurosciences, University of the Basque Country UPV/EHU, 48940 Leioa, Spain; 3Centro de Investigación Biomédica en Red Sobre Enfermedades Neurodegenerativas (CIBERNED), 28031 Madrid, Spain

**Keywords:** lipids, mitochondria, demyelination, dysmyelination, remyelination, axonal damage, neurodegeneration, repair, progressive multiple sclerosis

## Abstract

Multiple sclerosis (MS) is a complex autoimmune disease of the central nervous system (CNS), characterized by demyelination and neurodegeneration. Oligodendrocytes play a vital role in maintaining the integrity of myelin, the protective sheath around nerve fibres essential for efficient signal transmission. However, in MS, oligodendrocytes become dysfunctional, leading to myelin damage and axonal degeneration. Emerging evidence suggests that metabolic changes, including mitochondrial dysfunction and alterations in glucose and lipid metabolism, contribute significantly to the pathogenesis of MS. Mitochondrial dysfunction is observed in both immune cells and oligodendrocytes within the CNS of MS patients. Impaired mitochondrial function leads to energy deficits, affecting crucial processes such as impulse transmission and axonal transport, ultimately contributing to neurodegeneration. Moreover, mitochondrial dysfunction is linked to the generation of reactive oxygen species (ROS), exacerbating myelin damage and inflammation. Altered glucose metabolism affects the energy supply required for oligodendrocyte function and myelin synthesis. Dysregulated lipid metabolism results in changes to the composition of myelin, affecting its stability and integrity. Importantly, low levels of polyunsaturated fatty acids in MS are associated with upregulated lipid metabolism and enhanced glucose catabolism. Understanding the intricate relationship between these mechanisms is crucial for developing targeted therapies to preserve myelin and promote neurological recovery in individuals with MS. Addressing these metabolic aspects may offer new insights into potential therapeutic strategies to halt disease progression and improve the quality of life for MS patients.

## 1. What Is Multiple Sclerosis?

Multiple Sclerosis (MS) is a chronic and autoimmune disease of the Central Nervous System (CNS) that results in inflammation, demyelination, large focal lesions in the white and grey matter, and axonal loss [1]. The prevalence of the disease in both developed and developing countries is increasing exponentially and it usually occurs in young adulthood [2]. Furthermore, women seem to be two to three times more likely to be afflicted by the disease, related to an earlier onset but less progressive phenotype compared to men [3].

The current classification of MS encompasses four primary phases: clinically isolated syndrome, relapsing-remitting phase (RRMS), secondary progressive phase (SPMS), and primary progressive phase (PPMS). However, there is significant overlap among these stages, and the transitions between them remain unclear [4,5,6]. RRMS, which affects approximately 80% of patients, is characterized by inflammation, demyelination, and intermittent episodes of neurological dysfunction. Over time, untreated patients may progress to the SPMS phase due to its continuous worsening and reduced occurrence of clinical relapses. Conversely, around 10% of patients exhibit a PPMS phenotype from the outset, with no preceding clinical relapses [7]. This progression can result in escalating physical disability, cognitive impairment, or other neurological symptoms. Notably, a recent discovery known as “progression independent of relapse activity” (PIRA) has been observed in certain individuals with MS, where disease advancement occurs without any acute relapses typical of the relapsing-remitting phase. PIRA appears to be associated with chronic inflammatory tissue damage leading to substantial and early white matter atrophy [8,9].

From a histological point of view, focal lesions have been found in both white and grey matter, including the cortex, basal ganglia, brain stem, and spinal cord. The rupture of the Blood–Brain Barrier (BBB), the axonal loss, and the demyelinating lesions lead to substantial progressive neurodegeneration. Oligodendrocyte precursor cells and oligodendrocytes are widely affected by this disease [10]. When demyelination diffuses across tissue, microglia and inflammatory cells are recruited to clear the myelin debris and fight the hostile environment. There are two types of microglia present in MS brains, pro-inflammatory and anti-inflammatory microglia [1,11]. 

### Ongoing Basic and Clinical Research

The best way to reproduce the inflammatory aspect of MS is through the classical autoimmune mouse model known as Experimental Autoimmune Encephalitis (EAE). In this model, specific antigens or peptides are inserted to activate innate immune system and autoreactive T cells, resulting in diverse disease phenotypes and MS-like pathology [12,13].

The current strategy for treating MS relies on the use of disease-modifying therapies (DMTs) and symptomatic therapies. The neurological dysfunction observed in these patients is a fluctuating and disabling experience that can impact the quality of life and pose a great burden on the patients [14]. The treatment of DMTs consists of modulating the inflammatory component of MS on a more long-term basis, resulting in reduced relapse rates and disease progression. Many disease-modifying drugs have now been approved by the Food and Drug Administration (FDA), and there are still some in development [14]. Despite the advances in these drugs, they are often associated with mild or severe adverse effects and fail to prevent the axonal damage observed in the progressive form of the disease. DMTs can be further categorized into recombinant cytokines, complex peptide mixtures, monoclonal antibodies, and small molecules [15]. Over a dozen of DMTs have been approved for the treatment of MS, with each having a different mechanism of action. The approved drugs are the following [16,17]:Interferon beta medications: These synthetic injectable drugs mimic the naturally occurring protein interferon-β, which helps regulate the immune system and reduce inflammation.Glatiramer acetate: A synthetic mixture formulated with four different amino acids that replicates the structure of MBP. Upon its injection, it functions by redirecting the immune response away from myelin to prevent further damage.Teriflunomide: An oral medication that inhibits the proliferation of immune cells involved in MS through the inhibition of an enzyme called dihydroorotate dehydrogenase (DHODH).Dimethyl fumarate: An oral medication with anti-inflammatory and antioxidant properties that might exert its function by activating the Nrf2 pathway.Fingolimod: An oral medication acting as a sphingosine-1-phosphate receptor modulator. It traps immune cells in the lymph nodes, reducing their ability to attack the central nervous system.Natalizumab: An intravenous monoclonal antibody preventing immune cells from crossing the BBB and entering the central nervous system.Alemtuzumab: Intravenously infused monoclonal antibody that targets and depletes T and B cells involved in MS.Ocrelizumab: An intravenous monoclonal antibody specifically targeting and depleting B cells, mitigating the immune attack on myelin.Siponimod: An oral sphingosine-1-phosphate receptor modulator, similar to fingolimod. It prevents immune cells from reaching the CNS by trapping them in the lymph nodes.Cladribine: An oral medication primarily affecting lymphocytic function, leading to a reduction of certain immune cells.

In spite of the presence of numerous efficient and well-tolerated immunotherapies, a considerable proportion of individuals with multiple sclerosis (MS) still encounters severe and incapacitating symptoms. Although evidence-based symptomatic treatments are scarce, they still serve as a good strategy to alleviate MS-related symptoms and prevent secondary issues that may arise from them.

Remyelination is a key element of the neuroprotective and regenerative strategies designed to treat MS [18]. Myelin is an essential structure surrounding and protecting axons and it is also involved in metabolite exchange and support. Remyelination can be successful at the early stages of the disease due to the surviving pool of oligodendrocytes and their progenitors (OPCs). However, a higher neuron, oligodendrocyte, and OPC depletion can be observed as the disease progresses. The loss of these cell types and the inflammatory environment culminates in a failure to achieve remyelination [19]. There are currently various strategies at play to address the pathophysiology of MS, comprising diverse approaches such as remyelination, cell-based therapy, and metabolic regulation.

Remyelination techniques in the context of MS involve the use of various drugs and therapeutic approaches to stimulate the production of new myelin or support the repair of existing myelin. Numerous studies have identified various compounds that have the potential to enhance the recruitment, survival, and differentiation of oligodendrocytes, thereby leading to improved remyelination. Some of these medications target specific receptors in the brain, including muscarinic acetylcholine receptors (such as M1) and histamine receptors (such as H1 and H3). Examples of such drugs include benztropine, clemastine, quetiapine, ivermectin, and GSK239512 [20,21,22,23,24].

Additionally, Opicinumab is a promising target for remyelination, as it is a humanized monoclonal antibody specifically designed to counteract the effects of leucine-rich repeat neuronal protein 1 (LINGO-1). LINGO-1 is a glycoprotein found on the surface of both central nervous system (CNS) neurons and oligodendrocytes. Its presence inhibits various essential processes, including oligodendrocyte differentiation, myelination, neuronal survival, and axonal regeneration. Numerous in vitro and in vivo studies have indicated that blocking LINGO-1 can facilitate the remyelination of axons. However, clinical trials have shown mixed results [25,26,27]. 

Aside from employing drugs as therapeutic strategies, cell therapy offers several advantages in remyelination therapies. By introducing specific cells or stem cells that have the potential to differentiate into oligodendrocytes, cell therapy aims to directly repair the damaged myelin in the CNS. Additionally, cell-based therapies can exert immunomodulatory effects, reducing inflammation and providing a conducive environment for remyelination [22]. Presently, there are several ongoing clinical trials exploring the potential effects of these cells. Some trials aim to modulate inflammation, creating a conducive microenvironment for remyelination, while others focus on using OPCs to replenish endogenous niches or replace cells affected by genetic or age-related factors [28,29]. Promising cell types under investigation include mesenchymal stem cells and neural stem cells [30,31,32].

## 2. Oligodendroglia: Biology and Function

The CNS is a tight network that encloses specialized cells such as neurons and glia [33]. Up until the early 20th century, glia cells were defined as ‘Nervenkitt’ since, at the time, they were only thought of as glue that holds the neuronal tissue together. However, an exhaustive structural analysis of the brain by Pio del Rio-Hortega culminated in the description and differentiation of a subtype of cells known as neuroglia [34]. These cells were widely diffused along the white matter and grey matter. Unlike neurons, these cells were unable of growing their axons, however, they were composed of processes that made it possible for them to reach multiple axons [35]. Advances in microscopic and staining techniques categorized said neuroglia into three different cell types: astrocytes, microglia, and oligodendrocytes (OLs) [36]. The cells involved in the first stages of the oligodendroglia development had not yet been described due to a lack of suitable markers. In spite of that, Raff, Miller, and Noble made a significant discovery in 1983 by pinpointing the precursor cells responsible for oligodendrocytes (OPCs), thus establishing them as the fourth prominent glial subtype found within the brain [37,38].

Oligodendrocyte Precursor Cells: A Lookout into Their Functionality

During development, OPCs are the primary source for giving rise to fully mature and myelin generating OLs [39]. Despite their high proliferative capacity, OPCs only account for around 5–10% of the total cell population in the brain, with lesser amounts found in the grey matter [35,40]. OPCs are also known for their high characteristic expressions to the platelet-derived growth factor receptor alpha subunit (PDGFR-α) and NG2 markers, the latter being the reason why OPCs are also known as NG2 positive cells (Figure 1) [35,41]. 

During development, OPCs emerge from the ventricular zone of the brain and spinal cord where they will proliferate and migrate to different regions to further differentiate into oligodendrocytes. Although the peak of OPC proliferation occurs in the first postnatal stages, this rate declines with time [42]. Nonetheless, they continue to be one of the most actively dividing cell types in the adult central nervous system [43]. Adult OPCs have been described to hold the ability to generate myelinating OLs in the CNS, especially in demyelinating lesions [44]. Live imaging reveals that the loss of one OPC leads to swift division of a neighbouring OPC. This implies that all OPCs can enter the cell cycle, and lost cells are replaced not by mobilizing SVZ progenitors or a subset of highly proliferative OPCs, but rather through the collective homeostatic behaviour of the entire OPC population [45]. The differentiation of OPCs into immature OLs is driven by a series of extrinsic and intrinsic signals. Once OPCs have committed into a more mature phenotype, they lose their high proliferative capacity, rendering this step critical [46]. 

Notwithstanding the fact that OPCs are best known for being responsible of generating oligodendrocytes, they also perform multiple other functions in the brain [37]. Under homeostatic conditions, OPCs have achieved a more active contributor role. OPCs express multiple ion channels such as voltage-dependent Na+ channels and ionotropic glutamate and GABA receptors [47]. This renders OPCs the ability to receive synaptic input from neurons and maintain unique electrical properties, however, whether they are able to generate action potentials is still unknown [48]. Further studies have remarked the role that OPCs may have in supporting BBB integrity and mediating neuroinflammation, and in other functions [43,49,50]. Genetic ablation of NG2 positive OPCs not only resulted in reduced density and branching of the vascular network in the brain but also in microglial overactivation followed by neuronal death due to severe neuroinflammation [43].

b.Oligodendrocytes

Oligodendrocytes (OLs), a type of glia implicated in a wide array of neurological functions, are in charge of producing myelin sheaths around neuronal axons [39]. The differentiation step from OPCs to fully mature and myelin producing OLs is divided into four stages: proliferative OPCs, immature oligodendrocyte precursor cells (pre-OLs), differentiated OLs, and myelinating OLs [51]. During the developmental process, pre-OLs initiate the construction of myelin outgrowths around specific axons, leading to the loss of their bipolar morphology and transitioning into a more advanced, mature phenotype. At this stage, OLs show high expression for CNPase, O4 and O1 (Figure 1). Later on, mature oligodendrocytes will be able to produce multiple concentric layers of membrane around axons as well as myelin proteins. The Olig2 marker is a specific cell lineage marker for mature oligodendrocytes [34] and, together with the Sox10 transcription factor, unambiguously identify oligodendroglia lineage [52].

As mentioned before, oligodendrocytes are best known for their myelin generating capacity in the CNS white matter. The process of myelination varies depending on the brain region, with oligodendrocytes often insulating a significant number of axons. However, in certain instances, they may wrap only a single axon, as is the case in the ventral spinal cord [46]. The intricate interplay between axons and myelinating oligodendrocytes is supported by compelling evidence suggesting that ion balance is controlled by channels positioned at the interface between the myelin sheath and the axon [53]. As a widespread signalling molecule and second messenger, the regulation of intracellular calcium concentration ([Ca2+]i) within oligodendrocytes, including localized regions within myelin sheaths, has a direct impact on the process of myelin formation and remodelling, as well as potentially influencing other yet undiscovered functions [54]. Furthermore, oligodendrocytes undertake various non-canonical functions, including metabolic and trophic support to axons. One strategy to achieve this is through the release of lactate via the monocarboxylate transporter 1, which will be harnessed by axons to generate mitochondrial ATP [55].

## 3. Myelin in the CNS

At birth, humans possess a CNS that is primarily devoid of myelin, but following birth, there is a substantial increase in the population of oligodendrocytes, leading to extensive myelination during the early years of childhood. The process of myelination persists throughout adolescence and into adulthood, occurring in a distinct pattern across space and time, which corresponds to the development and preservation of functional neural circuits [56]. 

The presence of myelin sheaths facilitates the swift propagation of action potentials through a mechanism known as saltatory conduction. This is achieved by concentrating voltage-gated Na+ channels in small gaps between neighbouring myelin sheaths, known as the nodes of Ranvier, and by serving as electrical insulators. Axons that are fully myelinated along their entire length conduct impulses significantly faster compared to unmyelinated axons [53,57]. In addition, achieving adequate insulation relies on the specific types and proportions of lipids and proteins that constitute myelin.

The myelin membrane maintains essential characteristics of lipid bilayers present in cellular membranes, yet it possesses a unique structure and composition. Like other cell membranes, the myelin sheath consists of water, lipids, and protein molecules as its main constituents (Figure 2). However, the proportions of these components in myelin differ from what is typically observed in a regular cell membrane [58]. While the majority of biological membranes possess a balanced ratio of proteins to lipids, the myelin sheath has a higher proportion of lipids, constituting approximately 80% of its dry weight. This lipid-to-protein ratio in myelin plays a role in achieving the compact arrangement of the myelin sheath through non-covalent interactions between lipids and myelin proteins [59]. 

Myelin Composition: Lipids

The initial mapping of myelin membranes took place during the 1960s to 1970s. Although the precise lipidomic composition of myelin is still under investigation, an estimated ratio of approximately 2:2:1 has been proposed for cholesterol:phospholipids:glycolipids in myelin (Table 1) [60]. Except for cholesterol, fatty acids serve as essential building blocks for the majority of lipids found within the myelin membrane [61]. A comprehensive lipidomic analysis of myelin membranes has accounted approximately 700/600 different lipid species in both human and murine myelin. Among these lipids, around 60% could be readily categorized into established lipid classes such as phosphatidylcholines (PC), phosphatidylethanolamines (PE), sphingomyelins (SM), cerebrosides (Cer), sulfatides (Sulf), and others. However, the remaining 40% of lipids could not be assigned to any currently known lipid classes associated with myelin [62].

Myelin lipids play a crucial role in establishing the molecular interactions that enable myelin to attach firmly to the axonal membrane. Moreover, these lipids serve as a foundation for myelin proteins by forming microdomains known as lipid rafts [63]. The generation of these lipid rafts within the cell membrane play a crucial role in guiding membrane proteins to their specific destinations within the cell. Consequently, these lipid rafts have a significant impact on protein trafficking, signalling processes, as well as the fluidity and stability of myelin [64].

The maintenance of myelin membranes involves slow exocytic or endocytic processes that allow the replenishment and degradation of myelin constituents, a process known as myelin turnover [65]. During this process, myelin lipids undergo degradation within lysosomes, leading to their conversion into fatty acids. Subsequently, said fatty acids will be subjected to beta oxidation, taking place in mitochondria and peroxisomes. As a result, energy is generated and the necessary components for synthesizing new myelin lipids are produced [59]. Inhibition of oligodendroglial macroautophagy can disrupt the process of myelin turnover. Consequently, an age-dependent degeneration and accumulation of abnormal myelin structures have been observed [66].

**Table 1 ijms-24-12912-t001:** Approximate lipid composition of human and mouse myelin membrane.

Group	Overall Percentage
Cholesterol	40%
Phospholipid	38%
a. Phosphatidylcholine	7%
b. Phosphaditylethanolamine	7%
c. Plasmalogen	13%
d. Sphingomyelin	6%
b. Glycolipids	20%
b.1. Galactosylceramide	17%
b.2. Sulfatide	3%
Fatty acids	2%
a. Linoleic acid (18:2)	
b. Arachidic acid (20:0)	
c. Palmitic acid (16:0)	
d. Oleic acid (18:1)	
e. Eicosenoic acid (20:1)	
f. Arachidonic acid (20:4)	
g. Docosatetraenoic acid (22:4)	
h. Docosahexaenoic acid (22:6)	

Table modified from (Chrast et al., 2010; Gopalakrishnan et al., 2013; and Poitelon et al., 2020) [57,67,68]. (H = human/M = mouse).

b.Myelin Composition: Proteins

Proteins are primarily located within layers of the myelin membrane that wrap the nerve fibres, forming the insulating sheath. Despite their structural differences, these proteins have shared functions and act together to aid in the consolidation and compaction of the myelin membrane. The key proteins found in the myelin membrane include myelin basic protein (MBP), proteolipid protein (PLP), and 2′,3′-Cyclic nucleotide 3′-phosphodiesterase (CNPase), myelin oligodendrocyte glycoprotein (MOG), and myelin-associated glycoprotein (MAG) [69]. 

PLP is the predominant protein within the CNS myelin membrane, accounting for 38% of the total protein mass. The key role of PLP is to facilitate a compact multilayer membrane by attaching myelin lamellae together. Consequently, there will be an increase in the physical stability of myelin that will promote proper nerve conduction [58,70]. Following PLP, MBP remains the second most abundant CNS myelin protein, constituting approximately 30% of the overall protein mass. The shiverer mutant mice, which are deficient in the MBP gene, are unable to produce MBP proteins, and hence display incomplete myelin sheaths. This highlights the fundamental role of MBP as an essential structural protein in myelin formation [71,72].

## 4. Metabolic Profile of Oligodendroglia and Myelin: A Brief Insight into Lipid Metabolism

The CNS is especially vulnerable to metabolic changes since it is believed to lack energy stores [33]. Moreover, although the brain constitutes a small portion of body weight, its daily energy consumption oscillates around 20%. Out of this energy expenditure, neurons take up the majority since they need a vast amount of energy for rapid axonal impulse conduction [73]. 

Glucose, the main energy substrate used by brain cells, is involved in many essential functions such as ATP production, synthesis of neurotransmitters, and oxidative stress management [74]. Brain cells generate ATP from glucose mainly by two metabolic pathways: glycolysis in the cytosol and oxidative phosphorylation (OX-PHOS) in the mitochondria. ATP obtained from OX-PHOS is obtained in combination with the production of NADH from the citric acid cycle (TCA) [75].

OLs preferentially use glycolysis over OX-PHOS to cover their high ATP demand [76]. Under favourable and nutrient rich conditions, human derived OPCs and OLs rely significantly more on glycolysis to generate ATP. However, when faced with stress or pathological conditions, oligodendrocytes have the ability to withdraw their processes from the myelin sheath, adopting a less metabolic state as a survival mechanism. Nonetheless, this retraction results in the destabilization of compact axonal myelin sheath. In contrast, OPCs fail to retract their processes and therefore quickly undergo cell death under stress [77].

The elevated glycolytic activity of OLs can result in the generation of pyruvate in the cytosol, which serves as metabolic support for neuronal oxidative metabolism. Nevertheless, mitochondria require more time to utilize the generated pyruvate, resulting in an excess of pyruvate. This surplus pyruvate is converted into lactate, which is the final product of glycolysis [78]. Lactate is then released into the periaxonal space through the monocarboxylate transporter 1 (MCT1) and subsequently taken up by axons through the neuronal MCT2. This pathway by which oligodendrocytes provide trophic support to the axons is known as the oligodendrocyte–axon lactate shuttle [79,80]. The elevated lactate production and transport can have a positive impact on the axons. The myelin wrapping these axons create specialized channels known as “myelinic channels”, which facilitate the transport of lactate and other beneficial factors to the axons, thereby promoting the maintenance and health of the axonal structures [81].

In addition to serving as metabolic fuel for oligodendrocytes, lactate can also be used by these cells for lipid production. In vitro experiments have demonstrated that lactate can enhance the mouse OPC differentiation process [82]. When faced with low glycose availability or hypoglycemic conditions, the proliferative and differentiating capacity of oligodendrocytes is diminished, as well as an inhibition in their ability to produce myelin is introduced [83]. Consequently, it could be concluded that the proper functioning of the oligodendrocyte and axon lactate exchange is essential for the differentiation of cells originating from the oligodendrocyte lineage [84].

However, in the presence of oxygen, OLs are particularly resistant to glucose withdrawal [85], revealing alternative mitochondria-dependent energy sources. Drosophila hold the capability to switch towards breaking down FAs in mitochondria when there is scarcity of carbohydrates. Experiments conducted on Drosophila subjected to long term nutritional stress conditions have shown that glial cells switch from glycolysis to β oxidation to provide axons with nutrients. In addition, glucose deprivation and β oxidation inhibition in oligodendrocytes induced rapid neurodegeneration [86]. The findings highlight that glial cells can rely on the degradation of glial fatty acids or peripheral lipid stores, which can be converted into ketone bodies, to sustain brain function and ensure survival in unfavourable conditions. Furthermore, this also suggests that ketone bodies could serve as alternative metabolic fuel for neurons [86,87].

Since the 1960s, it has been recognized that the two primary ketone bodies, acetoacetate and β-hydroxybutyrate, serve as metabolic fuels for the brain [88]. However, their potential to replace glucose in neuronal oxidative metabolism has only been uncovered recently [86]. The liver plays a central role in producing and storing ketone bodies, utilizing acetyl-CoA derived from imported fatty acids for ketogenesis [89]. However, it is believed that local lipid stores within the cortex glia of Drosophila have the ability to utilize their own reserves to generate ketone bodies and transport them to areas that are experiencing nutritional deprivation. Therefore, in times of starvation, these newly synthesized ketone bodies can be taken up by neurons and used as energy substrates to sustain memory formation [90].

For constant myelination to occur, there needs to be a constant synthesis of lipids which are supplied by oligodendrocytes. Lipid metabolites can also serve as energy reserves when facing low glucose conditions [91]. The autophagy-lysosomal pathway is responsible for recycling myelin lipids which results in the release of fatty acids that can be used to produce new myelin lipids [69]. However, when glucose availability is insufficient, OLs can redirect the fatty acids released during myelin breakdown towards β-oxidation resulting in acetyl-CoA production which acts as a substrate for OX-PHOS. Experiments conducted on a mouse optic nerve model have demonstrated that generating ATP from lipids allows for a greater allocation of glucose-derived metabolites to the axons, thereby supporting their preservation [78].

Peroxisomes, small organelles present in the cytosol of most eukaryotic cells, are also involved in a series of functions involving metabolism and detoxification of reactive oxygen species. Like mitochondria, they are essential for the β-oxidation of fatty acids, especially of very long chain fatty acids (VLCFA) [92] (Kassmann et al., 2007). Peroxisomes are abundant and can be spotted in numerous CNS cells, especially in glial cells. Recent studies have demonstrated the presence of peroxisomes within the innermost tongue of the myelin sheath [93,94]. The inhibition of peroxins, the proteins involved in the biosynthesis of these organelles, leads to their dysregulation and, consequently, to white matter abnormalities [95]. Thus, following these results, peroxisomes have been hypothesized to have a significant role in direct axonal support and myelin maintenance. Furthermore, peroxisomal inhibition studies have concluded that peroxisomes from myelin forming glia perform β-oxidation to chain shorten VLCFAs for their subsequent transfer into axonal mitochondria [94,96]. 

## 5. Altered Bioenergetics in Multiple Sclerosis

Several research studies employing various biological samples have now provided compelling evidence of abnormalities in the metabolome among individuals with MS [97,98,99,100]. Cerebrospinal fluid (CSF) samples of MS patients have revealed increased levels of lactate whilst also showing disturbances in glucose and energy metabolism. Studies using mass spectrometry-based metabolomics have shown alterations in lipid and fatty acid metabolism, with elevated concentrations of circulating free fatty acids and products of fatty acid oxidation being observed [101,102].

Mitochondrial Dysfunction and Oxidative Stress

Two hypotheses have been proposed to explain the origin and pathological features of MS. The first one aims to explain that the cascade of events begins in the periphery where dysregulated auto-reactive T cells enter the CNS along with macrophages and B cells. This starts a target-directed attack on myelin sheaths, resulting in a RRMS course and CNS injury. Conversely, the second hypothesis states that MS is a neurodegenerative disease, with the initial malfunction or auto-inflammatory behaviour occurring within the CNS itself. In this hypothesis, chronic shedding of antigenic cell components occurs, leading to an inflammatory response and subsequent degeneration of the myelin sheath [21,102].

The second hypothesis, which proposes that MS is a neurodegenerative disease, finds support in the possibility of mitochondrial dysfunction. This dysfunction could lead to an energy deficiency and hinder various processes such as impulse transmission, axonal transport, and ion trafficking [103]. Evidence from an EAE study shows that this occurs before neurological dysfunction becomes apparent [104]. Furthermore, the impaired mitochondrial function not only leads to energy deficits but also triggers the activation of inflammatory cells within the CNS, further propelling the progression of neurodegeneration [105,106]. MS patients are characterized by reduced ATP production, potentially resulting from decreased activities of the mitochondrial electron transport chain complexes [107]. However, it is worth noting that in certain instances, complex V activity is sometimes enhanced as a compensatory mechanism for the lower activity of complex I. Despite this compensatory effort, the disruption in mitochondrial function ultimately results in decreased energy production [108]. 

Besides its involvement in altered bioenergetics, mitochondrial dysfunction is strongly associated with oxygen metabolism and the generation of Reactive Oxygen Species (ROS). As the brain utilizes a substantial proportion of inhaled air (20%) and consumes a significant amount of tissue oxygen (90%) for energy production, this metabolic process leads to the production of detrimental ROS [109]. ROS and their reactive products attack all classes of biomolecules, including lipids. Notably, the CNS, which comprises membranes with elevated levels of polyunsaturated fatty acids, renders neuronal cells highly sensitive and vulnerable to damage caused by the adverse effects of these reactive species. The consequence of the interaction between ROS and nitric oxide is the generation of highly reactive peroxynitrite [105]. This by-product specifically endangers OPCs due to their limited antioxidant defense mechanisms. As a result, OPCs are unable to undergo maturation into myelin-forming oligodendrocytes, leading to impaired myelin production and maintenance in the context of MS [110].

b.Inflammation and Bioenergetics Interplay

Mononuclear phagocytes (MPs) act as the CNS’s surveillance system, primarily responsible for local immune surveillance. The resident MPs include microglia, constituting approximately 5–10% of all brain cells, and macrophages residing in the perivascular spaces and choroid plexus [111]. Neurodegeneration results in a shift in the balance of microglial activation towards their pro-inflammatory state. This leads to the production of chemokines/cytokines, such as tumour necrosis factor-alpha (TNF-α), interleukin (IL)-6, IL-1β, and IL-12, which have pro-inflammatory properties [105].

Simultaneously, the secretion of these pro-inflammatory cytokines has adverse effects on crucial mitochondrial components, leading to a decline in mitochondrial respiratory chain function and exacerbating neurodegeneration [112]. Additionally, the enzymes responsible for the tricarboxylic acid cycle and oxidative phosphorylation within the mitochondria are adversely affected in the inflamed CNS due to the presence of these proinflammatory mediators [113]. Although the evidence suggests that inflammatory factors impact mitochondrial dynamics, the specific molecular mechanisms through which these mediators induce damage is still poorly understood.

Moreover, macrophages and microglia are recognized for their role in clearing myelin [23]. Nevertheless, during inflammatory conditions and when they adopt their pro-inflammatory state, they lose this capability, leading to increased oxidative stress due to disrupted myelin clearance [114]. Myelin also serves as a crucial barrier against oxidative stress. However, in the context of multiple sclerosis (MS), the disruption of myelin clearance leads to the accumulation of damaged myelin, exposing nerve fibres to increased oxidative stress causing a breakdown in nerve signal transmission and leading to neurological symptoms characteristic of MS [115].

It is essential to comprehend the intricate relationship between inflammation and bioenergetics in MS to explore potential therapeutic approaches. By addressing mitochondrial dysfunction and metabolic changes in immune cells, novel treatment possibilities could emerge, leading to disease progression slowdown. 

c.Glucose Metabolism in MS

In MS, there is evidence of altered glucose metabolism, with reduced glucose uptake and utilization in certain regions of the brain. This metabolic dysfunction can impact the energy supply to neurons and other cells, potentially contributing to neurodegeneration. Glucose deprivation conditions are believed to have detrimental conditions on OPCs, resulting in fewer and thinner processes while oligodendrocytes tend to shift to a more glycolytic state for their survival [116].

Furthermore, the lactate levels in both the serum and CSF of MS patients vary depending on the clinical stage. During the early stages of the pathogenesis, lactate levels tend to be lower, but as the disease progresses, an increase in lactate expression can be observed. It can be concluded that there is an elevation in extra-mitochondrial glucose metabolism in MS patients, which could also be associated with impaired mitochondrial function [117]. Taking into account the aforementioned data, the fluctuations in lactate levels have the potential to serve as diagnostic criteria as they could indicate the progression of the disease [118].

d.Dysregulation of Lipid Metabolism in MS

Lipids play vital roles in the brain, participating in various processes, including neurogenesis, signal transduction, neuronal communication, membrane compartmentalization, and the modulation of gene expression. Because of their essential structural and physiological functions, any changes in lipid metabolism may indicate abnormal brain function [119]. Scientific evidence indicates that there is a modification in the lipid metabolism of the arachidonic acid pathway, which undergoes changes in the context of multiple sclerosis pathology [120]. Furthermore, specific lipid abnormalities have been reported, such as deficiencies in FA 18:2 and FA 20:4, as well as total PUFA (polyunsaturated fatty acids), accompanied by compensatory increases in saturated fatty acids with shorter carbon chains [121].

Utilizing the untargeted lipidomics approach, studies have revealed distinctive lipid signatures in multiple sclerosis (MS) patients compared to healthy controls. MS patients exhibit a unique phospholipidomic profile, characterized by significant reductions in key phospholipids, including phosphatidylethanolamine (PE) and phosphatidylcholine (PC) species, which play crucial roles in antioxidant functions. Therefore, certain phospholipids hold promise as potential biomarkers for clinical applications in the context of multiple sclerosis (MS) [122,123,124]. 

Macroautophagy and lysosome-mediated degradation are essential processes that contribute significantly to myelin turnover and regeneration. Nevertheless, pathological and aging conditions frequently lead to a decline in the efficacy of these mechanisms. Consequently, this disruption negatively impacts the overall well-being of oligodendrocytes, impairing their ability to synthesize fatty acids crucial for myelin maintenance and energy production [69].

In addition, sphingolipids, important components of lipid bilayers with functional and structural roles, have been implicated in MS disease processes [125]. Recent research highlights alterations in sphingolipid pathways that could contribute to oligodendrocyte injury, suggesting dysregulated anti-inflammatory and pro-inflammatory lipids as potential contributors to MS pathology. Ceramide lipids, such as sphingosine, is associated with oligodendrocyte damage and acute demyelination [126]. Ceramide release from stressed oligodendrocytes could trigger autoimmune responses following active demyelination, potentially serving as a diagnostic and prognostic marker [127].

Studies also link cholesterol derivatives, known as oxysterols, to inflammatory demyelination in MS. With about 25% of the brain consisting of cholesterol, higher than in other organs, oxysterols, particularly 24(S)-hydroxycholesterol, can impact CNS cells due to disrupted BBB [128,129]. Oxysterols influence lipid synthesis by affecting transcription factors, such as sterol regulatory element-binding proteins (SREBPs), which regulate genes involved in lipid homeostasis [127]. 

Additionally, dysregulations in lipid metabolism can profoundly affect the stability and integrity of the myelin sheath due to a disruption in the interplay between lipid receptors and myelin proteins [130]. These molecular players orchestrate pivotal processes within the myelin sheath, including the intricate machinery driving its synthesis and restoration. By disturbing their functions, aberrant lipid metabolism could potentially impede the myelin sheath’s ability to maintain its structural integrity and engage in reparative activities [131]. Moreover, MBP, aside from its arrangement and behavior within the myelin sheath, also influences the organization of the lipid bilayer within myelin. Thus, alterations in MBP dynamics might thus reverberate throughout the lipid environment, potentially contributing to the overall stability and functionality of the myelin sheath [73,132].

All in all, considering the complex interplay of bioenergetics and lipid metabolism, the exact mechanisms underpinning these interactions remain complex and call for deeper investigation. More extensive research is crucial for unravelling whether the bioenergetics irregularities observed in MS specifically affect certain groups of lipids, as opposed to influencing the entire range. This would open possibilities to target specific lipid types whilst sparing others. Such findings could pave the way for targeted interventions aimed at specific lipid categories while leaving others unaffected.

## 6. A Tentative Hypothesis on How Altered Bioenergetics May Affect MS Progression

Bioenergetics can significantly impact MS progression by impairing the function of the axon–myelin unit [133]. Indeed, disruption of the axon–myelin unit can occur acutely, such as during transient ischaemia, or chronically in MS and other neurodegenerative diseases, and involve myelinic NMDA receptors, a key component regulating the supply of energy substrates between oligodendrocyte/myelin and axons [134]. Thus, glutamate released by axons during action potential propagation activates NMDA receptors in oligodendrocytes that translocate glucose transporter GLUT1 into the plasma membrane, promotes glycolysis, and favours lactate shuttling to axons. This mechanism fine-tunes axonal energy demands during neuron-to-neuron communication, and it is impaired during dysmyelination and demyelination, leading to mitochondrial dysfunction and, eventually, to axonal damage. The detailed mechanisms would include: (i) energy failure due to the inability of oligodendrocytes to generate lactate for export to the axon, with a reduction on ATP synthesis in axonal mitochondria; (ii) an ensuing failure of ion transporters to maintain Na^+^ and K^+^ and therefore, impaired action potential propagation; and (iii) axonal Ca^2+^ dyshomeostasis causing enhanced activation of calpains, phospholipases, and other enzymes that ultimately result in structural axonal damage. 

Accordingly, demyelinated axons in focal plaques in remitting-relapsing MS may undergo transient energy deficits compromising their function (signal transmission) and structure. Partial remyelination in shadow plaques may restore energy supply and prevent the fatal fate of compromised axons, but not in acute and chronic plaques without overt remyelination. Consequently, long-term bioenergetic deficits can lead to axon demise, anterograde and retrograde neuronal degeneration, and MS progression. 

## 7. Concluding Paragraph

In this review, we provide an insight into the metabolic profile of oligodendrocytes and myelin, as well as in myelin lipid metabolism, along with a summarized account of how and where metabolic alterations occur in MS. This includes energy deficits in axons with the consequent slower signal propagation, oligodendrocyte dysfunction in rebuilding myelin after demyelination, and in lipid metabolism, all of which inexorably lead to MS progression.

Targeting those bioenergetic pathways and metabolic dysfunction in MS may offer new opportunities for interventions aimed at preserving neuronal function, promoting remyelination, and slowing the progression of MS. As myelin is the earliest compartment impaired in this disease, therapeutic strategies at MS onset and its progressive phases should be focused on preserving myelin. In this regard, the potential of myelin boosters promoting myelin build-up and adjusting lipid metabolism may contribute to CNS energy support.

## Figures and Tables

**Figure 1 ijms-24-12912-f001:**
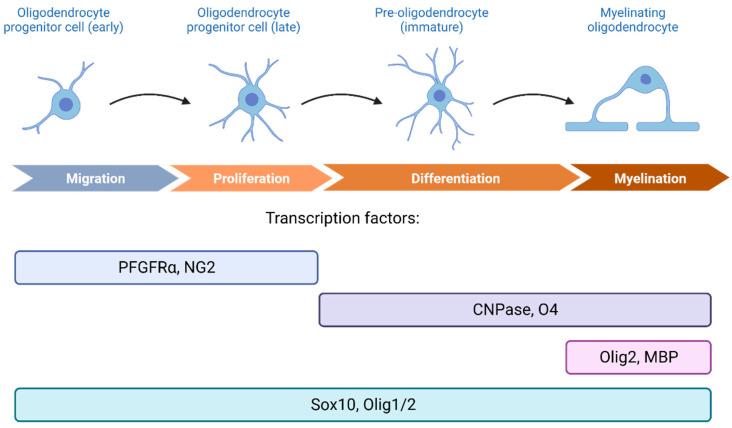
A visual depiction of the various stages involved in oligodendrocyte differentiation and maturation, along with their distinctive markers. (Created with Biorender).

**Figure 2 ijms-24-12912-f002:**
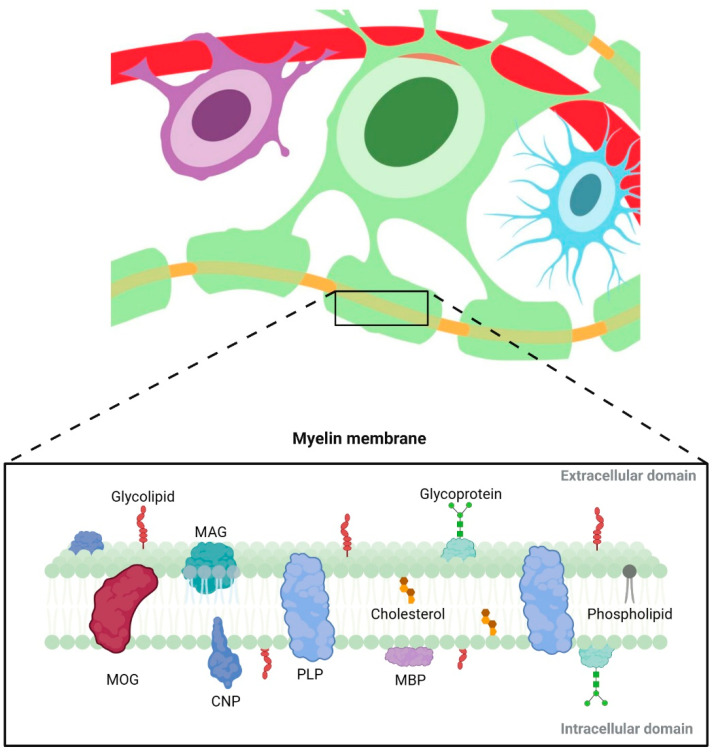
The structure and molecular makeup of the myelin membrane in the CNS. (Created with Biorender).

## Data Availability

No new data were created.

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
