# Peer review of "Alterations of Oligodendrocyte and Myelin Energy Metabolism in Multiple Sclerosis"

_ijms, 2023, doi:10.3390/ijms241612912_

Round 1
Reviewer 1 Report
The review article by López-Muguruza et al., is an interesting article for the readers. It gives an overview of the mechanism of oligodendrocyte and myelin energy metabolism in MS.
However, there are few issues that need to be addressed.
1. First page, second paragraph, authors have talked about 4 phases but have just mentioned 3 clinical forms of MS. Please use a recent classification of MS and also introduce the recent concept of PIRA in MS.
2. In Ongoing basic and clinical research section. Please mention all disease modifying therapies and symptomatic therapies.
3. Figure 2, needs to be put in bigger fonts or magnified.
4. Table 1, please remove the column of species as in all the cases authors are mentioning about humans and mice. This column is not giving any additional information.
5. Please check in the whole article that full forms with abbreviations should be only mentioned once in the beginning not repeatedly like Reactive Oxygen Species (ROS), Relapsing-remitting multiple sclerosis (RRMS) etc.
Author Response
Referee 1 (highlighted in blue in the revised manuscript):
- Clinical Classification and PIRA Concept: We have enhanced the manuscript by incorporating an updated clinical classification of Multiple Sclerosis. Additionally, we have introduced the concept of PIRA (Progression Independent of Relapse Activity) to enrich the contextual understanding of the study's background and significance. This augmentation serves to provide readers with a more comprehensive framework.
- Therapeutic Approaches: In response to the reviewer's request, we have conducted a comprehensive review to include the principal disease-modifying and symptomatic therapies utilized for enhancing the quality of life and mitigating symptomatology in Multiple Sclerosis patients.
- Figure Enhancement: To optimize reader comprehension and prevent any confusion, we have refined Figure 2 to improve the legibility of the accompanying labels.
- Table and Abbreviations: We have duly incorporated the suggested modifications to Table 1. Furthermore, throughout the article, we have meticulously addressed the reviewer's concern by minimizing the repetition of full-form abbreviations.
Reviewer 2 Report
The review offers an extensive and comprehensive report on how metabolism affects MS progression. There are some minor points to be addressed before the publication of the paper.
The authors should conduct a thorough proof reading of the text. In section 1 they present a sub-title marked “a. Ongoing basic and clinical research”. There is no part b, thus, it would be ideal to change the title of the section to incorporate this aspect.
Furthermore, the EAE model should be stated that it is a mouse model. In this section the authors talk about therapies such as remyelination. It would be advantageous to the paper for the authors to offer a very short comment on how these strategies are implemented and what types (e.g. peptides or non-peptides) of drugs are implemented.
In the following statement “...CNS in the presence or absence of insults...” what do the authors mean by insults? Do they talk about specific neurotransmitters or other chemical signals? This should be clarified.
In sections of the paper the authors employ phrases such as “studies show…” and the respective reference is only one study. If there are more than one study then the authors should provide an additional 2 or 3 references. Otherwise, these phrases should be edited.
Since the role of bioenergetics is poorly understood, it would be ideal for the authors to add a small commentary with their hypothesis on how these aspects may affect MS progression. The concluding paragraph should be extended summarizing the most important aspects of the review and provide a comprehensive conclusion based on the data provided.
One aspect of the paper is that the authors extensively describe myelin composition in section 3 but when talking about the lipid metabolism and how it affects MS progression there is no information on whether these abnormalities in bioenergetics affect certain lipids more or whether it the abnormalities affect only some lipids. Additionally, it would be ideal to discuss, if there is any information on the topic, on how these dysfunctions in lipid metabolism affect the proteins of the myelin sheath.
Author Response
Referee 2 (highlighted in yellow in the revised manuscript):
- Subtitle Alignment: In accordance with the reviewer's suggestion, we have adjusted the subtitle "Ongoing Basic and Clinical Research" to harmonize with the text content.
- Comprehensive Therapeutic Overview: In addition to briefly describing disease-modifying and symptomatic therapies, we have meticulously outlined the existing remyelination strategies employed for treating multiple sclerosis. We have elaborated on various drugs targeting muscarinic or histamine receptors, as well as the innovative cell therapy options currently undergoing clinical trials.
- Enhanced Explanation: We have provided a more detailed explanation of demyelinating conditions in place of the term "insults," expanding on how these conditions influence the homeostatic state of oligodendrocyte precursor cells (OPCs) in adulthood, thereby restricting their differentiation potential.
- References and Evidence Strengthening: Where statements were based on single studies and introduced with phrases like "studies show," we have reinforced the evidence base by either incorporating additional references or revising the wording for improved accuracy.
- Hypothesis Inclusion and Conclusion Expansion: A new section has been introduced proposing a preliminary hypothesis on the potential impact of altered bioenergetics on MS progression. Additionally, the conclusion section has been expanded to comprehensively summarize the key aspects addressed in the review.
- Lipid Metabolism Insights: While acknowledging the nascent nature of research on lipid metabolism impairment in MS, we have supplemented the manuscript with insights into how these bioenergetic abnormalities potentially affect specific lipid pathways. The manuscript now underscores the ongoing inquiry into whether these abnormalities target the complete lipidome or operate in a more focused manner. Additionally, we have introduced a brief exploration of the influence of these aberrations on myelin proteins and the consequent impact on the structural integrity of the myelin sheath.